# Quasicrystal nucleation and ℤ module twin growth in an intermetallic glass-forming system

Wolfgang Hornfeck[1], Raphael Kobold[2], Matthias Kolbe[2], Matthias Conrad[3] & Dieter Herlach[2]

While quasicrystals possess long-range orientational order they lack translation periodicity. Considerable advancements in the elucidation of their structures and formative principles contrast with comparatively uncharted interrelations, as studies bridging the spatial scales from atoms to the macroscale are scarce. Here, we report on the homogeneous nucleation of a single quasicrystalline seed from the undercooled melt of glass-forming NiZr and its continuous growth into a tenfold twinned dendritic microstructure. Observing a series of crystallization events on electrostatically levitated NiZr confirms homogeneous nucleation. Mapping the microstructure with electron backscatter diffraction suggests a unique, distortion-free structure merging a common structure type of binary alloys with a spiral growth mechanism resembling phyllotaxis. A general geometric description, relating all atomic loci, observed by atomic resolution electron microscopy, to a pentagonal ℤ module, explains how the seed's decagonal long-range orientational order is conserved throughout the symmetry breaking steps of twinning and dendritic growth.

[1] Institute of Physics, Academy of Sciences of the Czech Republic, 18221 Prague, Czech Republic. [2] Institute of Materials Physics in Space, German Aerospace Center (DLR), 51170 Cologne, Germany. [3] Department of Chemistry, Philipps University, 35032 Marburg, Germany. These authors contributed equally: Wolfgang Hornfeck, Raphael Kobold, Matthias Kolbe. Correspondence and requests for materials should be addressed to R.K. (email: raphael.kobold@dlr.de)

A first account relating a dendritic growth morphology of a crystal to its structure is Kepler's 1611 treatise On the Six-Cornered Snowflake[1] studying the closest sphere packings in 2D and 3D, distinguishing the (face-centered) cubic (fcc) from the hexagonal one (hcp). Explaining most of the crystal structures of metals, an atom is surrounded by 12 nearest neighbors in a cuboctahedral (fcc) or anti-cuboctahedral (hcp) shell. Yet another mode of twelvefold coordination, and locally more dense, is achieved by the icosahedron, one of the Platonic solids known since antiquity. However, its rotational symmetry is not compatible with lattice translations. Thus, it features in pioneering theories about the structure of liquids, as well as in Frank's explanation[2] of Turnbull's empirical studies[3] on the undercoolability of metallic melts: High undercoolings are possible when the local structures of the solid and the melt differ substantially. Then, a structural reorganization prior to solidification is required, acting as a barrier for nucleation[4,5]. Non-crystallographic, yet dense packing of spheres into icosahedral shells were envisaged by Mackay[6], who also explored what he called a pentagonal snowflake[7]—a crystallographic realization of Penrose's tiling of forbidden pentagonal symmetry—predicting its diffraction pattern[8] prior to Shechtman's paradigm-shifting discovery of quasicrystals[9,10]. A fierce controversy about their existence, in opposition to traditional explanations based on multiple twinning put forward by Pauling, was settled by awarding Shechtman the 2011 Nobel prize for chemistry. In a l'esprit de l'escalier we propose a nucleation and growth mechanism intimately connecting a quasicrystalline seed with a multiply twinned microstructure in what may be called a decagonal snowflake.

The discovery of quasicrystals challenged our understanding of order at the atomic scale. Structurally complex, yet crystalline intermetallics and metallic glasses represent competing states of condensed matter among metallic phases. Binary Ni–Zr alloys, in particular, form complex crystal structures and, upon rapid solidification, metallic glasses, including the congruently melting line compound NiZr[11,12], while in the ternary system with Ti an icosahedral quasicrystal exists[13]. This makes Ni–Zr a perfect system for studying the interplay between these distinct states of condensed matter—complex intermetallics[14], amorphous alloys and glasses[15,16], and quasicrystals[17–19]—and systematic investigations into dendritic growth in a chemically simple, compositionally well-defined metallic alloy. Non-equilibrium solidification experiments of NiZr were performed in an undercooling regime bounded by its melting and glass transition ($T_m = 1533$ K, $T_g \approx 730$ K[20]; cf. the Supplementary Discussion and Supplementary Table 1).

## Results

**Electrostatic levitation (ESL) of NiZr.** ESL is a state-of-the-art method for containerless processing of materials maintaining ultra-high vacuum purity conditions. A processing cycle includes the laser-melting and (under-)cooling of the sample, until spontaneous crystallization occurs at the nucleation temperature $T_n$. Locally released latent heat yields an emissivity contrast observable with a high-speed camera (HSC). A sketch of the experimental setup is shown as Supplementary Fig. 1. For NiZr the intersection of the growing crystal's convex hull with the sample's spherical surface shows a decagonally shaped solidification front moving at an average velocity of $\langle v \rangle_{[001]} = 0.62(4)$ m/s at the maximum achieved undercooling $\Delta T_{max} = T_m - T_{n,min} = 300(5)$ K (Fig. 1).

**Microstructure of NiZr solidified at maximum undercooling.** Microstructural analysis includes optical polarization and scanning electron microscopy (SEM), with electron backscatter diffraction (EBSD) mapping the crystallographic orientation of intergrown grains by means of false-color maps and inverse pole figure (IPF) plots (Fig. 2). As-solidified, spherical samples, overcasted with 10 longitudinal geodesics converging at two poles, marking the onset point of solidification and its antipode. Series of perpendicular cross-sections confirm the alignment of the polar axis with the rotation axis of a tenfold microtwin (TMT), in which the twin domains are separated by coherent large-angle grain boundaries traversing the sample with an angular inclination of about 36°. Each domain exhibits a coarsened dendritic fine structure, in which coherent and incoherent, straight and wavy grain boundaries alternate, distinguishing their arrangement from all-parallel straight boundary deformation twins. Coherent boundaries constitute the stem of every dendrite (Supplementary Fig. 2), with twinned dendrites visible up to the third-order reflecting a spatial hierarchy of structural organization, originating in perfect registry from a single center: the primordial nucleus.

**Statistical proof of homogeneous nucleation in NiZr.** The case for homogeneous nucleation is made by the thermodynamical and kinetic parameters determined from the statistical analysis of a series of 200 consecutive, Poisson-distributed nucleation events (see Supplementary Fig. 3, Supplementary Table 2, and the Supplementary Methods sections for details of the analysis)[21]. The (dimensionless) solid–liquid interfacial energies $\alpha = 0.5949$ and $\sigma = 0.2107(6)$ J m$^{-2}$ match the values for metallic systems characterized by polytetrahedral/icosahedral order[5]. The pre-exponential factor $K_V = 1.034 \times 10^{35}$ m$^{-3}$ s$^{-1}$ of the nucleation rate compares well to Turnbull's estimate of $\sim 10^{39}$ for the homogeneous nucleation in pure metals[3]. The skew distribution of nucleation events, with a steep decrease at higher relative undercoolings $\Delta T / T_m$, as well as the large undercoolings achieved at small variation, unequivocally point to homogeneous nucleation as its causative effect.

Now, with the experimental observations going from top to bottom reaching their natural end at the scale of atoms, we proceed to show how all of the macroscopic features emerge from the crystal structure of NiZr and its geometric peculiarities going from bottom to top via geometric model building.

**Crystal structure of NiZr.** NiZr crystallizes in a CrB-type structure comprising eight atoms in a $C$-centered, orthorhombic unit cell of space group $Cmcm$, $a = 326.8(8)$ pm, $b = 993.7(4)$ pm, $c = 410.1(5)$ pm. Both Ni and Zr occupy the Wyckoff site 4c (0, $y$, 1/4) with $y_{Ni} = 0.0817(17)$ and $y_{Zr} = 0.3609(8)$[22]. Now, the axial ratio $a/b$ of the NiZr unit cell causes the ($\pm$110) pair of net planes to enclose an angle of $2 \tan^{-1}(a/b) \sim 36.4°$, almost the tenth of a full circle. This simple yet crucial metrical relation is a prerequisite for the iris-like matching of 10 wedge-shaped twin domains around a common center.

As a caveat, one should note, that the solidification behavior of NiZr is more complicated if one considers the whole range of undercooling values. For undercoolings below 70 K NiZr primarily nucleates with the CsCl-type structure. Between 70 and 250 K a competition can be observed for NiZr solidifying in either the CsCl-type ($\approx$20% of the cases, see ref. [23] for an experiment performed at $\Delta T \approx 200$ K) or the CrB-type. Above 250 K CrB-type NiZr is the primary nucleating phase—exclusively (a Supplementary Discussion extends on this topic, see also Supplementary Figs. 4–7).

**Properties of the twin boundary.** The symmetry operation between adjacent twin domains is not a pure reflection but consists of a non-vanishing translational component (shift) $\sigma$[24].

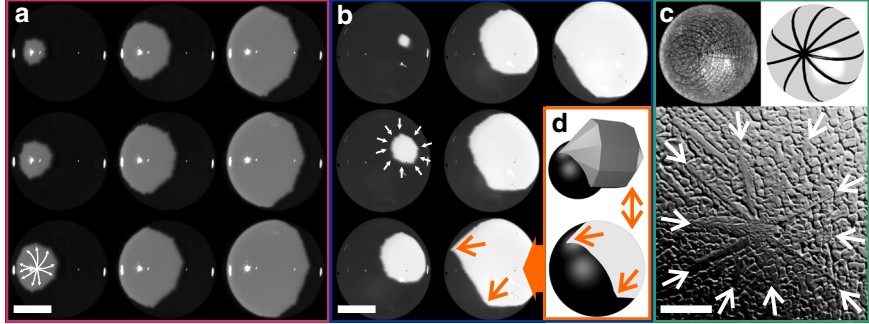

**Fig. 1** Decagonal solidification of deeply undercooled NiZr. HSC snapshots of decagon-shaped solidification fronts for two samples of NiZr (red and blue **a**, **b**, respectively, scale bar 1 mm). An SEM micrograph (green **c**, scale bar 50 μm) shows a close-up of the geodesic surface structure of the levitated droplet. A simulation (orange **d**) illustrates the shape of the solidification front as the intersection of the growing crystal, an elongated decagonal bipyramid, with a sphere

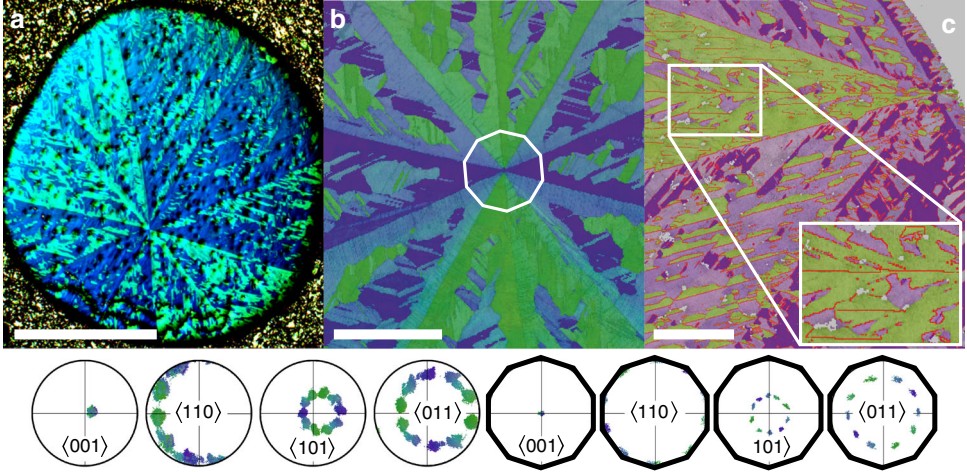

**Fig. 2** Tenfold twinned NiZr. Dendritic microstructure of NiZr cross-sections (**a** (scale bar 500 μm): optical polarization micrograph; **b** (scale bar 100 μm) and **c** (scale bar 200 μm): false-color EBSD-maps) is confirmed by a series of IPFs (bottom): A single cluster of crystal orientations aligns with the axial growth direction along a common [001]-axis of the NiZr twin domains, while a perfect tenfold clustering is observed for perpendicular ⟨110⟩-directions. Circular IPFs represent the crystallographic orientations for the complete EBSD-map of a slightly inclined sample; decagonal, alignment-corrected IPFs highlight perfect orientational relationships for the highlighted central part of the microtwin

However, assuming a shift with rational components of (1/4)[110][24] does not result in an optimal twin boundary. We found that an irrational shift yields a perfect match, if a minor idealization on the atomic coordinates is made, in conjunction with an out-of-plane shift of $(1/2)[001]$[25] accounting for a distance optimizing polar domain structure. The independent specification of the unit cell metric ($b/a = \sqrt{5 + 2\sqrt{5}} \sim 3.078$), the irrational shift $\sigma[110] + (1/2)[001]$ ($\sigma = \pm(4\tau)^{-1} \sim \pm 0.155$), and the atomic coordinates ($y_{Ni} = \tau^{-2}/(2\sqrt{5}) \sim 0.085$ and $y_{Zr} = \tau/(2\sqrt{5}) \sim 0.362$), where $\tau = (1 + \sqrt{5})/2$ represents the golden ratio, yields a unique structural model, in which, quite remarkably, the structure in the bulk is identical to the structure across the twin boundary (Fig. 3a)! This exceptional feature of distortion-free, energetically advantageous twin boundaries explains their dominant occurrence as a macroscopic growth feature. Notably, the CrB-type itself is a periodic unit-cell twin of the *fcc*-type[26].

**A geometric model for twin growth.** Non-vanishing, same sign shifts around a common center induce a chiral arrangement of atoms into spirals forming alternating shells of a facetted cyclic intergrowth[27] structure (Fig. 3b). For our choice of shift each site $n$ in a spiral is determined by an explicit expression $(\lfloor\sqrt{2(n-1)}\rfloor + \mathbf{1}_M(n))$ mod 10, in which the set $M =$

$\{\lceil 2(m + \sqrt{m})\rceil | m \in \mathbb{N}\}$, representing a sequence of integers ranging between 0 and 9 and consecutively denoting the geometrical locus of sites with respect to a decagonal vector basis. Here $\lfloor \cdot \rfloor$, $\lceil \cdot \rceil$, mod, and $\mathbf{1}_M(n)$ are the number theoretic floor, ceiling, modulo, and indicator functions, respectively. Only steps of ±36° occur locally, traced out in a turtle graphics-like manner. Spiral growth in the **ab**-plane extends along simple geometric rules into a twin pattern maintaining the long-range orientational order of the quasicrystalline nucleus. Upon dendrite formation a cooperative growth sets in, conserving the twin boundaries into the dendrite stems, ultimately yielding the geodetic pattern of surface twin faults. Probing the atomic structure with X-ray diffraction (XRD) and high-angle annular dark field scanning transmission electron microscopy (HAADF-STEM) confirms the spatially correlated microtwinning in the bulk (Fig. 3c), as well as the irrational shift of $\sigma = \pm(4\tau)^{-1}$ at the twin boundary (Fig. 3d).

Growth along the **c**-direction extends the Zr-centered pentagonal antiprism of Ni atoms, first into an compressed icosahedron with two Zr atoms at its apices and eventually into a pentagonal antiprismatic column, a structural motif well-known for many intermetallics. Breaking the icosahedral symmetry the compression distinguishes one out of six fivefold symmetry axes for growth along the tenfold axis, favoring a $(2+1)$D axial growth model, in which the in-plane spiral growth mechanism is

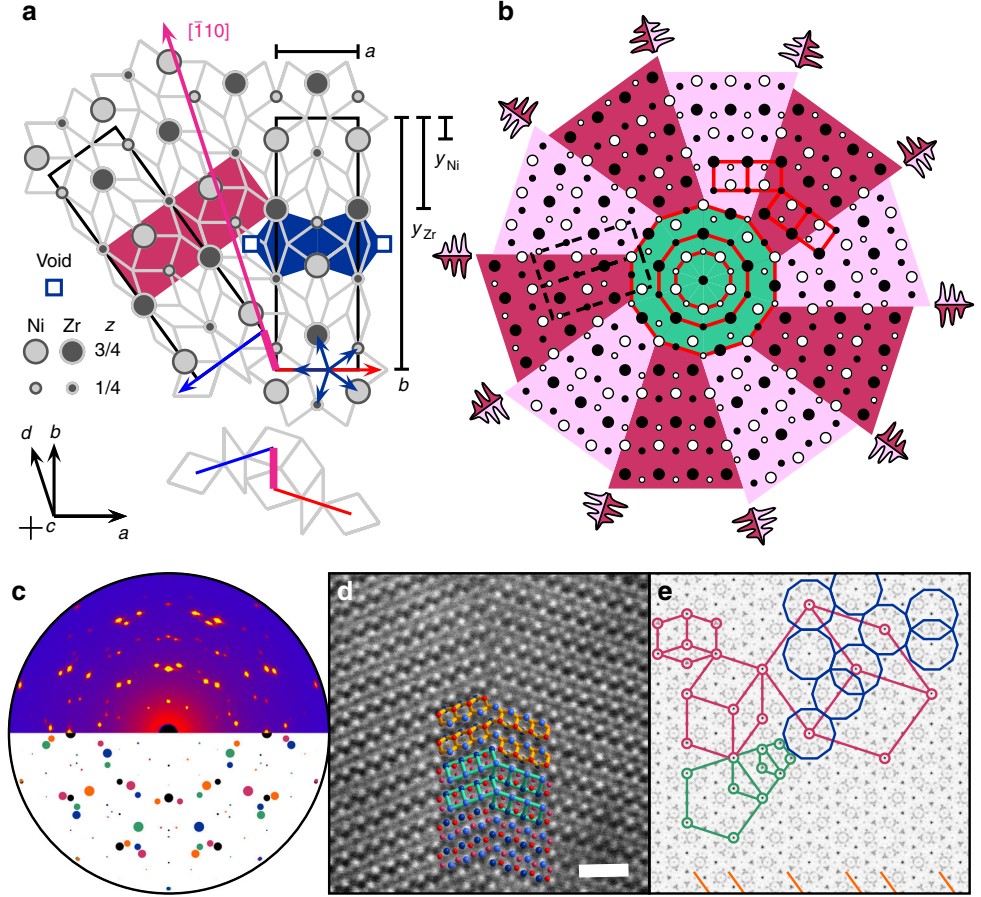

**Fig. 3** Atomic structure of the tenfold twin. **a** Coherent twin boundary constructed from a rhombic Penrose tiling, its vertices described by a pentagonal $\mathbb{Z}$ module (dark blue star of arrows). A pair of defective pentagons (dark blue) contains formal defects, not to be confused with structural vacancies (!), being attributed to geometric frustration regarding the alternating height of atoms in the $c$-direction. Structural units (dark red) are coherent across the twin boundary (magenta). **b** Patch of the tenfold twin with its chiral and polar domain structure (light and dark red), showing coherent structural motifs across the twin boundary (solid red and dashed black), and its quasicrystalline core (green), with $\tau$ scaled decagons (red). The schematic onset points of dendrite growth are highlighted. **c** Measured (top) and calculated (bottom) X-ray diffraction pattern. **d** HAADF-STEM micrograph of a coherent twin boundary and its overlayed structure model (Ni light/dark red, Zr light/dark blue, interatomic contacts yellow and tortoise, scale bar 1 nm). **e** Fourier transform of the point pattern in **b** exhibiting relations to the rhombic Penrose tiling (red), pentagonal inflation symmetry (green), decagonal cluster coverings (blue), golden ratio spacings (orange)—tell-tale features of long-range orientational order associated with decagonal quasicrystals

amended with an out-of-plane columnar one (Supplementary Fig. 8). Preferred atomic attachment sites are found both in the re-entrant kinks of adjacent twin domains and along the staggered stacking of pentagons into antiprisms. A single compressed icosahedron suffices to trigger growth into the observed macroscopic twins. Indeed, while TMTs nucleate already in abundance during the heating of splat-cooled[24] or melt-spun[28] amorphous NiZr, it is only via ESL, undisturbed by convection, that a single TMT emerges from the undercooled melt with the radial extension of its major twin boundaries solely limited by the sample's diameter. In contrast, a twinning mechanism based on regular icosahedra, in which dopants ensure the formation of a quasicrystalline seed, enhances the heterogeneous nucleation of a competing *fcc*-type phase, yielding grain-refined samples[29,30].

**Relations to quasicrystals.** Tenfold microtwinning in NiZr is fully correlated: Except for a natural scaling factor due to the atomic sizes no free parameter is involved. A reason for this is given by the novel concept of $\mathbb{Z}$ module twinning[31,32], in which twinning does not interrelate two or more lattices, but $\mathbb{Z}$

modules as their mathematical generalization instead. The case of NiZr, as its first explored realization, accounts to a twinning of a pentagonal $\mathbb{Z}$ module, which is a projection of the 5D primitive hypercubic lattice along its hyperspace diagonal [11111] onto 2D real space. Similar projection schemes are well-known in the realm of quasicrystals[10], highlighting the conceptual similarity between the NiZr TMT and decagonal quasicrystals. Indeed, the $\mathbb{Z}$ module strictly couples all atomic positions, with their projected coordinates $(x, y)$ determined by five integers $a$–$e$ ($\phi = 2\pi/5$): $(x, y) = \sqrt{2/5}\,(a + (b + e)\cos(\phi) + (c + d)\cos(2\phi),\ (b - e)\sin(\phi) + (c - d)\sin(2\phi))$[31,32]. A corresponding vertex-decorated tiling of thin and thick Penrose rhombuses features Ni and Zr atoms arranged into motifs of defective pentagons in all possible rotated and reflected orientations throughout the twin.

Accordingly, a 2D Fourier map of a finite patch of the underlying point pattern shows distinctive features of long-range decagonal orientational order beyond a mere tenfold radial symmetry (Fig. 3e). The pattern's innermost core exhibits a $1{:}\tau{:}\tau^2$ scaling for three successive shells. The transition from this quasicrystalline frozen-in seed of nucleation to the

microtwin's domain structure is marked by a loss of higher-order scalings at a distance of ~0.7 nm comparing well with the critical radius $r^* = 1.1$ nm estimated from classical nucleation theory.

Generalizing these geometrical considerations the 132 known binary CrB-type structures can be classified according to their axial ratios $b/a$ and $c/a$ (Supplementary Fig. 9 and Supplementary Tables 3 and 4). In a twin description, all structures exhibit a compressed central icosahedron, however, with the geometrical flexibility and potential to form $n$-fold twins for $7 < n < 12$. Similar microtwins exist for ternary substitution variants of NiZr with Cu and Hf, as well as for binary, CrB-type AlZr, CoZr, and NiB (Supplementary Table 5).

NiZr offers unique insights into the genesis of a dendritic microstructure in a glass-forming alloy, based on the combination of fundamental crystallography with advanced yet generally applicable processing and characterization techniques. A general geometric mechanism bridges the scales from a homogeneously nucleated quasicrystalline seed via twinning-induced growth to a macroscopic solid. Most comprehensively, our ideas seem suited to model microstructural quasicrystal–(twin)crystal transitions for axial quasicrystals with even-fold rotational symmetry ($n = 8$, 10, 12, … ; Supplementary Fig. 10), addressing specific aspects regarding the outstanding question of how such entities might grow[33], and extending to the generic case of a dense, binary sphere packing, with potential applications in materials design and soft matter science.

NiZr, as a missing link connecting quasicrystals and multiple twins, illuminates intermediate states of order between amorphous alloys, crystals and their twins, and quasicrystals.

## Methods

**Sample preparation**. Spherical reguli (2.5 mm in diameter, 70 mg) were prepared by arc-melting stoichiometric mixtures of high-purity metals (Ni: 99.999%; Zr: 99.97%) in a Ti-gettered Ar atmosphere. Mass losses during arc melting/ESL were <0.1 mg with nominal compositions checked on embedded/polished samples by energy dispersive X-ray spectrometry.

**Electrostatic levitation**. ESL was performed with a custom-built levitator using an infrared laser ($P = 75$ W, $\lambda = 808$ nm) and an ultrafast pyrometer (Impac IGA 120-TV). A Photron Ultima APX 775k HSC allowed the direct observation of the solidification with 10,000 fps and ($512 \times 320$) px time and space resolution. In particular, every nucleation event out of the 200 ones used for the nucleation statistics, has been checked by visual inspection of the HSC recordings for the presence of the decagonal solidification front before inclusion in the nucleation statistics dataset.

**Electron backscatter diffraction**. EBSD samples were embedded in a carbon containing, electrical conductive phenolic mounting resin via application of temperature and pressure (450 K, 25 kN). Cylindrical resin specimens were grinded and polished (0.05 μm colloidal alumina) with a final treatment using a slightly basic suspension (0.1 μm colloidal silica, pH = 9.0). The success of the oriented embedding was examined using an optical polarization microscope (Imager.A2M, Zeiss). A LEO 1530VP SEM ($U_{acc} = 20$ kV, 1 nm spatial resolution) equipped with an EDS-system (INCA) and an EBSD detector (Oxford HKL) was used to characterize the microstructure in terms of phase composition and crystal orientation.

**X-ray diffraction**. XRD of a tenfold twinned specimen was measured on a STOE IPDS II at 293 K using Mo-$K_\alpha$ radiation to a $2\theta_{max}$ of 62°.

**Electron microscopy**. HAADF-STEM was performed on a $C_S$-corrected FEI Titan Themis 200 microscope operated at 200 kV (Schottky XFEG cathode, 0.08 nm point resolution; image conditions: 17.6 mrad probe (half) convergence angle with 70 pA probe current, 69–200 mrad annular dark field detection angle). Final sample preparation included twin-jet electropolishing with 10% perchloric acid and 90% methanol at 243 K and 35 V and ion milling in a GATAN PIPS.

## Data availability

All relevant data is available from the corresponding author upon reasonable request.

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

## Acknowledgements

The authors thank D. Holland-Moritz as well as M. Quiquandon and D. Gratias for fruitful discussions, P. Kuhn for comments on the manuscript and the German Science Foundation (DFG) for financial support (HE1601/28). W.H. received financial support by Czech Science Foundation (GACR) grant 18-10438S. We kindly acknowledge H. Rösner, L. Kienle, and U. Schürmann for preliminary HRTEM investigations. HRTEM and HAADF-STEM experiments, providing the image of Fig. 3d, have been kindly performed by F. Mompiou, G. Patriarche, A. Sirindil, D. Lamirault, and S. Lartigue-Korinek, of which a detailed study will be published elsewhere. We also thank F. Kargl for his continuing endorsement of the research on NiZr.

## Author contributions

D.H. initiated the research and took part in the discussion of the results, R.K., M.K., and W.H. designed, conducted, discussed, and analyzed the major part of the experiments (synthesis, ESL, HSC, nucleation statistics, optical and scanning electron microscopy, EBSD, fabrication of pre-thinned, oriented specimens for HRTEM sample preparation), and devised the atomistic model. M.C. performed the X-ray diffraction experiment. W.H. and R.K. wrote the article.

## Additional information

**Competing interests:** The authors declare no competing interests.

