## [Peer Review File · Nature Communications]

Reviewers' comments:

Reviewer #1 (Remarks to the Author):

After a hiatus, the revised manuscript has made some very significant edits and additional data. In particular, the AC-STEM clearly shows the atomistic model for the twin formation is consistent with their model. However, science moves on. The authors didn't address a 2015 paper in the text, although they do mention it in passim in the supplementary section by D.G. Quirinale, et al. Appearance of metastable B2 phase during solidification of Ni₅₀Zr₅₀ alloy: electrostatic levitation and molecular dynamics simulation studies. *J. Phys.: Condens. Matter* 27 (2015) 085004 (6pp). Using a similar ESL approach, but with in situ XRD they show that the initial nucleating phase is the B2. Note that fig. 10 shows a similar transient in the region between finishing of the recalescence and the radiant cooling of the sample. What the paper didn't address is - could the twinned growth be a mechanism by which the CrB-type structure grows is not from the melt, but a solid state manifestation of the growth off the B2? This twin growth maybe a mechanism by which the large strains and chemical ordering be accommodated? Many of their geometric arguments can be applicable to solid state nucleation rather than from the liquid only.

This possibly needs to be addressed in the new manuscript.

Reviewer #2 (Remarks to the Author):

The authors have thoroughly revised the paper. In particular, they have added X-ray diffraction and high-resolution STEM data. They also clarified a number of important points raised by the referees.

I am satisfied that the paper is publishable in its present form. Since it is an interesting "afterthought" on the old structural controversy following the discovery of quasicrystals, it would be of interest to a broad readership of *Nature*. I recommend its publication.

Reviewer #3 (Remarks to the Author):

This is a piece of interesting work on the growth of 10-fold twinned dendritic crystals in an undercooled NiZr melt by an electrostatic levitation experiment. The careful geometric analysis provides a new insight into the formation of 10-fold twins in a CrB-type NiZr alloy. Both electrostatic levitation experiments and geometric analysis presented in the paper are very impressive. However, I cannot support the paper for publication in the current form for the following reasons:

1. The key conclusions of the paper are too speculated. The authors did not provide any direct evidence to prove the existence of a quasicrystalline seed which leads to the growth of tenfold twinned dendrites. Technically, it is not difficult to image the quasicrystal seeds, if they exist, by transmission electron microscopy. This does not require a sub-angstrom spatial resolution from an aberration-corrected TEM. The combination of conventional high resolution TEM with a focus ion beam system for slicing a thin TEM sample from the center of the tenfold dendrites should be enough for acquiring the direct evidence. I don't understand why the authors did not take any action to prove since the paper was submitted to *Nature* in 2014. Frankly, without the piece of solid evidence of the existence of quasicrystalline seeds, all the conclusions and discussion in the

paper are groundless.

2. The structural model in Fig. 3b does not tell that the quasicrystalline seed is indispensable for the tenfold twin formation. One can simply extrapolate the boundaries of the tenfold twin boundaries (actually they are not twin boundary and will be discussed later) to the center of the quasicrystalline core without changing any atom positions. Only a simple icosahedron cluster is sufficient to coordinate the tenfold twins. Thus, geometrically, a quasicrystalline seed is not essential for the formation of the tenfold twined dendrites.

3. Page 3 line 1, the authors claim $T_g=730$ K. I am curious to know how the authors determined the T_g of the ZrNi alloy. The authors claimed "NiZr, as a missing link connecting quasicrystals and multiple twins, illuminates intermediate states of order between glasses, crystals and their twins, and quasicrystals". In fact, as pointed out by the previous reviewer #3, the binary NiZr alloys are not metallic glass former. Rapid quenching by melt spinning and PVD can make amorphous NiZr alloys but those alloys do not have detectable glass to supercooled liquid transition before crystallization. Technically, it is not possible to measure the T_g . Moreover, since NiZr is not a glass former, all the discussions on the inherent relation of the tenfold twins, quasicrystals with glasses are baseless.

4. The so-called ten-fold twins are neither real twins, nor have coherent interfaces, as evidenced by the authors' HRTEM image and model. The mirror-symmetry has been broken at the dendrite interfaces. Technically, the EBSD cannot tell the tenfold dendrite is a "distortion-free structure" because of the poor resolutions of SEM-EBSD in both real space and reciprocal space.

5. The homogeneous nucleation by a single seed is derived from the statistical analysis of about 200 observations. However, the authors did not clarify whether all the 200 nucleation events are by the tenfold dendrites. From Figure 1, one can see that all the nucleation events presented in the paper appear to initiate on the surface of the droplet(s). Can the preferred surface nucleation still give rise to the homogeneous nucleation?

Consequently, I am not convinced by the revised manuscript and the response letter. The paper gives me the feeling that the authors stretched too much from the results told by their experiments. The key conclusion cannot stand without the TEM image verifying the existence of the quasicrystalline cores. Thus, I cannot support the paper for publication in the current form.

Remarks on the referee comments

Referee comments	Response
Referee #1	
After a hiatus, the revised manuscript has made some very significant edits and additional data. In particular, the AC-STEM clearly shows the atomistic model for the twin formation is consistent with their model. However, science moves on. The authors didn't address a 2015 paper in the text, although they do mention it in passim in the supplementary section by D.G. Quirinale, et al. Appearance of metastable B2 phase during solidification of Ni₅₀Zr₅₀ alloy: electrostatic levitation and molecular dynamics simulation studies. J. Phys.: Condens. Matter 27 (2015) 085004 (6pp).	The paper of Quirinale et al. is an important contribution, and we thank the referee for highlighting the apparent contradiction with our work, but it does not tell the whole story, because the solidification behaviour of NiZr is much more complicated in detail. We have included the reference in our manuscript with a detailed discussion given in the supplemental material to the article. It should be noted, that a paper of Wilson and Mendeleev, which is also concerned with the calculation of interfacial energies, is contradicting the results of Quirinale et al. in parts, especially concerning the primary nucleating phase, as well as the relative stabilities of B2 and B33 phase. We think, this topic is not yet settled, but would say that our study could be seen as a complementary piece of the mosaic and certainly add value to the discussion of this topic.
Using a similar ESL approach, but with in situ XRD they show that the initial nucleating phase is the B2.	This is true, but only under certain circumstances, mainly depending on the undercooling temperature. It is not true, however, for the case presented in the current version of the manuscript, i.e. NiZr at highest undercoolings. Both cases can be discerned experimentally by their distinctive T-t diagrams (as well as by their growth velocities, solidification morphology, microstructure). If the B2 phase (CsCl type) is the first phase to nucleate, there is an additional thermal signal (recalescence) present, while this is missing from cycles in which the B33 (CrB type) phase is the first to nucleate. We have performed extensive experiments to clarify this topic, including synchrotron X-ray phase selection

	measurements and think while Quirinale et al.'s measurements show the situation at 200 K undercooling, they do not tell anything about the situation at highest undercoolings.
Note that fig. 10 shows a similar transient in the region between finishing of the recalescence and the radiant cooling of the sample.	The transient might seem similar on the first glance, but the details of the thermal signal, i.e. its peak shape, are entirely different. Moreover our own experiments show that the solidification of the B2 phase is always accompanied by its own recalescence, i.e. primary B2 and secondary B33 solidification is a two recalescence process in our case, while only one (for the B2 solidification) is seen in theirs. Moreover, they write: "There is a spike in the thermal data which appears to coincide with the initial appearance of B33, and is consistently observed in all runs; however, it is uncertain whether this spike represents the signature of an actual physical process, or is simply an artifact of the measurement technique and subsequent analysis". It should be noted that we have performed more experiments on $\text{Ni}_{50+x}\text{Zr}_{50-x}$ and $\text{Ni}(\text{Hf},\text{Zr})$ which show that the position of the spike is dependent on the composition of the samples and thus indeed represents a physical process, although it is so far unclear, what exactly causes it so reproducibly.
What the paper din't address is - could the twinned growth be a mechanism by which the CrB-type structure grows is not from the melt, but a solid state manifestation of the growth off the B2?	While this might be a possible mechanism to envisage, we follow Occam's razor and prefer our, much simpler atomistic mechanism. To be clear, we do not rule out that there could be a crystallographic, especially metrical relationship between the CsCl- and CrB-structure type, which would favor a mechanism envisaged by the referee. But with cubic symmetry in the core one would rather expect a different kind of microstructure, especially a more isotropic one, with either 6, 8, or 12 (or multiples thereof, say 48 as the highest order of a cubic point group) as the dominant multiplicity. But in any case we studied, we see a (2+1)D microstructure, with 10+1 directions, which clearly favours our model. It should also be noted that our claims are quite specific. If anyone can add some detail,

	they should feel free to do so, but the mere possibility of (much more complicated) alternating theories should not be detrimental to the publishing of our work, which we see as contributing to a discussion about the topic, and not as some dogma no one ever will be able to doubt. Alternating theories of course would also need some experimental proof. So far, however, there is no such proof, but plenty of indications that can be explained by our model and our model alone.
This twin growth maybe a mechanism by which the large strains and chemical ordering be accommodated?	It is our conviction that the presented twin growth mechanism based on the selective, primary solidification of the CrB type phase explains that there are neither reasons for the build-up of large strains at all (due to the perfect matching domain boundaries) nor difficulties in the accomodation of the chemical ordering (due to the spiral growth). Both ideas highlight that this complicated twin structure can grow as simple and regularly as a single crystal, where no one expects large strains or a lack of chemical ordering. Of course, the question of large strains and chemical ordering would arise, if the twin structure would be resulting from a B2 type phase nucleus, but exactly because no such difficulties arise with the proposed CrB type phase, why should one choose for a less plausible model?
Many of their geometric arguments can be applicable to solid state nucleation rather than from the liquid only.	This is certainly true, and covers a theoretically most interesting question, but practically almost impossible to prove. Even if the B2 phase would be the first to nucleate in all circumstances, that must not mean that it grows as fast as the second phase to nucleate. Cases of this are know in the eutectic systems Ni-Sn and Co-Si, where a competition exists between the first-to-nucleate, slowly growing phase and the second-to-nucleate, fast growing phase, with the latter dominating the final microstructure. Here are some references: Yang et al. Acta Mater. 59 (2011) 3915; Zhang et al. Acta Mater. 61 (2013) 4861; Wang et al. Acta Mater. 142 (2018) 172.

	It should be noted that under any circumstances we are analyzing the microstructure after the cooling. Our conclusions have to be based on indirect proof, much like in the reconstruction of a crime scene, because no one is able, not even with in situ X-ray measurements, to really tell what happens when and how. Regarding this challenge, however, we think we can offer a fully self-consistent picture of NiZr's solidification which matches all our experimental evidence.
This possibly needs to be addressed in the new manuscript.	A full account of the aforementioned details of the complicated solidification behaviour of NiZr is given in the PhD thesis of Raphael Kobold, published in english, which can be obtained online at: https://hss-opus.ub.ruhr-uni-bochum.de/opus4/frontdoor/index/index/docId/4938. In addition to citing this link also in the supplemental material we have added an extensive discussion there too, together with a short note in the revised manuscript referring to it. We chose to do so due to the length restrictions of the main manuscript and because we think this topic is not the main point of our paper. We hope this is sufficient to clarify this point both for the referee and any future reader of our article.
Referee #3	
This is a piece of interesting work on the growth of 10-fold twinned dendritic crystals in an undercooled NiZr melt by an electrostatic levitation experiment. The careful geometric analysis provides a new insight into the formation of 10-fold twins in a CrB-type NiZr alloy. Both electrostatic levitation experiments and geometric analysis presented in the paper are very impressive. However, I cannot support the paper for publication in the current form for the following reasons:	
1. The key conclusions of the paper are too speculated. The authors did not provide any direct evidence to prove the existence of a quasicrystalline seed which leads to the growth of tenfold twinned dendrites.	1) The quasicrystalline seed is a direct and inevitable consequence of the proposed twin model. It joins the central icosahedron with the peripheral CrB type fragments. Without it, the whole model breaks apart disconnected.

	2) Now, this twin model is solely determined by the relative shift of the twin domains at their boundaries. There is no other free parameter to adjust. 3) We proved the occurrence of this shift by the atomic resolution HAADF-STEM picture. Therefore, in a logical conclusion, from 1) to 3) we proved the necessary existence of the decagonal seed, too. Moreover, the FFT of a patch of the model exhibits all kind of features observable not in any kind of twins perceivable but only known from quasicrystals (in particular, decagonal long-range order). The Z module construction allows to project down the twin structure in the same way as one can perform it for quasicrystals. We agree that our model does not describe a part of a known bulk decagonal quasicrystal occurring as the seed, if that was expected. It would be anyhow difficult to state this for a patch of structure with only a few nanometers of spatial extension. We use the term quasicrystal for describing and highlighting the structural features closely related to quasicrystals found in the structure.
Technically, it is not difficult to image the quasicrystal seeds, if they exist, by transmission electron microscopy.	We can only state that four electron microscopy groups in Germany and France did not find it simple to image NiZr samples with the TEM for the last three years. Moreover, to be precise, it is one seed that one has to find, because other than Kuo we do not have a multitude of them scattered in an amorphous matrix but just one right in the center that is the important one to analyze. And even Kuo did not manage to find a tenfold seed at highest resolution, that is one with all ten domains meeting at one center and with perfect imaging conditions for all ten domains simultaneously fulfilled (which means perfect orientation of the sample).

This does not require a sub-angstrom spatial resolution from an aberration-corrected TEM. The combination of conventional high resolution TEM with a focus ion beam system for slicing a thin TEM sample from the center of the tenfold dendrites should be enough for acquiring the direct evidence.	Well, it does, because the quasicrystalline seed in question is not bigger than a few nanometers in extension. We do know about FIB and PIPS and all kind of other ways of preparing TEM samples, and we did use them, too. In the case of NiZr it is not about throwing in some tiny piece of substance in a random fashion and look at it for five minutes. We had to orient and cut our samples with low tolerance, pre-thin them, and these were the simple steps. Thinning NiZr to electron transparency has proven to be non-trivial and difficult enough. Getting a twin boundary inside the thinned region is hard, but manageable, as we show. Getting the center just with the right offset to the thinning region is more luck than anything else.
I don't understand why the authors did not take any action to prove since the paper was submitted to Nature in 2014.	We have engaged electron microscopists from four different groups in Germany and France to achieve a TEM investigation with state-of-the-art atomic resolution, of course including the attempt to visualize the central core of the twin structure.
Frankly, without the piece of solid evidence of the existence of quasicrystalline seeds, all the conclusions and discussion in the paper are groundless.	It should be noted that similar quasicrystal seeding mechanisms exist in the literature (we cite them), that deliver no direct experimental proof at all for their claims, because they claim that even the quasicrystalline seed is vanished in these microstructures, due to a postulated phase transition into the observable phase. We think one has to compare these claims with ours and with respect to the challenge of "seeing" a seed by any kind of direct observation in general (even in situ experiments would fail to observe a single seed due to its inevitably small size).
2.The structural model in Fig. 3b does not tell that the quasicrystalline seed is indispensable for the tenfold twin formation.	It does. Because without this intermediate part of the structural model, the model is disconnected, and does not make any sense at all.
One can simply extrapolate the boundaries of the tenfold twin boundaries (actually they are not twin boundary and will be discussed later) to the center of the quasicrystalline core without changing any atom positions.	The referee suggests that we should do, what we already did. The decagonal core is the result of an extrapolation of the twin boundaries to the center.
Only a simple icosahedron cluster is sufficient to coordinate the tenfold twins.	No, it isn't. Of course the simple icosahedron cluster is forming the core, and in this way is

	more important for the coordination of the ten twin domains, but, as our model shows, there is a clear intermediate region present, neither being pure icosahedral, nor of the crystalline CrB type structure, which has to be spoken about in some way. By the way, a regular icosahedron would fail the symmetry breaking to a (10+1)-orientational microstructure, which our irregular icosahedron facilitates with ease.
Thus, geometrically, a quasicrystalline seed is not essential for the formation of the tenfold twined dendrites.	This is an ad hoc claim without any argumentative backing. It is essential. Geometrically and physically. Geometrically, because without it, that is solely based on a single icosahedron, no similar twin structure can be constructed. Furthermore it is physically necessary. A single icosahedron is far too small to nucleate anything, while the decagonal core of our model has just the right size for the critical radius of a nucleus as demanded by classical nucleation theory!
3. Page 3 line 1, the authors claim $T_g=730$ K. I am curious to know how the authors determined the T_g of the ZrNi alloy.	We did not determine it. We never claimed. We took this value from the literature, and as far as the referee highlights by his comment, that we missed to give a citation to that value, we thank him.
The authors claimed “NiZr, as a missing link connecting quasicrystals and multiple twins, illuminates intermediate states of order between glasses, crystals and their twins, and quasicrystals”. In fact, as pointed out by the previous reviewer #3, the binary NiZr alloys are not metallic glass former.	We grant that the glass forming ability (GFA) of NiZr is a matter of controversy, but mostly regarding its quantitative degree and not its qualitative presence. Certainly eutectic NiZr alloys are better known for their GFA. However, while different authors both include and exclude NiZr from the glass forming range of the binary system, there are several reports stating a measured glass transition temperature (T_g) for NiZr, too. If someone measured a T_g, it is qualitatively a glass for us. We do not say that NiZr is a good glass-former. It isn't. But that's a different story. And anyhow, since there exist better glass-formers in the same binary system, neighboring NiZr, it is not too far fetched, in our opinion, to make a connection to them.

	We adjusted the title to "glass-forming system" in order to highlight this fact and ameliorate our claim in this regard. We also on some occasions replaced glass by amorphous alloy, when we made statements about NiZr (without doubt Kuo produced amorphous NiZr). Moreover, we added a discussion focused solely on that topic in the supplemental material, citing almost all available sources on the GFA of NiZr, of which an overwhelming majority treats NiZr in its equiatomic composition as a (marginal) glass-former.
Rapid quenching by melt spinning and PVD can make amorphous NiZr alloys but those alloys do not have detectable glass to supercooled liquid transition before crystallization.	It is well known and discussed in the literature, that NiZr poses a difficulty or even impossibility in measuring the T_g with DSC methods. That does not mean that one cannot measure it with a different method (as the cited people did).
Technically, it is not possible to measure the T_g.	Well, Myung et al. did it! Not by DSC, granted, but by 'thermomechanical analysis under continuous heating as a function of applied stress and heating rate' (Myung et al., Mat. Sci. Eng. A 179/180 (1994) 252-255). Moreover, Buschow states glass temperatures for almost equiatomic alloys and even shows a DSC curve for equiatomic NiZr, from which he extrapolated the glass transition point (Buschow, J. Phys. F: Met. Phys. 14 (1984) 593-607). Finally, two very recent MD simulations find NiZr amorphous (see references S22 and S23 in the supplement).
Moreover, since NiZr is not a glass former, all the discussions on the inherent relation of the tenfold twins, quasicrystals with glasses are baseless.	We disagree for the reasons stated before.
4. The so-called ten-fold twins are neither real twins, nor have coherent interfaces, as evidenced by the authors' HRTEM image and model.	The referee is right, that our twin structure does not follow the strict definition of twinning as put forward in the International Tables of Crystallography, which states that, colloquially speaking, twin symmetry operations do not contain any translational part (we guess that this is the point the referee is unhappy about, we don't know it for sure). It should nevertheless be clear, that all previous scientists spoke of twins, when they analyzed the case of NiZr. The structure is not

	just a random intergrowth, it shares all the regularity commonly known from twins. Moreover, the concept of Z module twins generalizes the twin definition to cases where strict twinning is happening in higher dimensions, while the down-projected real space structures might exhibit additional translational symmetry in their twin operations (see the cited work of Gratias for this extension of a twin definition to include many more cases than previous ones do).
The mirror-symmetry has been broken at the dendrite interfaces.	Yes! This is an integral part of our model and it ultimately creates a chiral structure, which is one of the intriguing features of our model. We do not mind that this might interfere with a conservative notion of twinning prone to be replaced by a more general definition, like that of Gratias.
Technically, the EBSD cannot tell the tenfold dendrite is a “distortion-free structure” because of the poor resolutions of SEM-EBSD in both real space and reciprocal space.	We developed an ideal model explaining the twin structure at the atomic level. Of course we extrapolate to the EBSD scale of micrometers under the assumption that the same model holds there too. Because it is plausible that it does. This is perfectly scientific reasoning.
5. The homogeneous nucleation by a single seed is derived from the statistical analysis of about 200 observations. However, the authors did not clarify whether all the 200 nucleation events are by the tenfold dendrites.	Every cycle was checked with high-speed camera observation. We made an annotation to this in the manuscript's experimental part.
From Figure 1, one can see that all the nucleation events presented in the paper appear to initiate on the surface of the droplet(s).	Well, it appears like that, and it is possibly true that most nucleation events in levitated melt droplets occur near to the surface, for reasons given in the next comment. However, nucleation events deep in the bulk are also observable. It might not be emerge from the manuscript but one can deduce the approximate point of nucleation from a geometrical analysis and simulations concerning the degree of simultaneousness of the appearance of the surface heat front on the samples.
Can the preferred surface nucleation still give rise to the homogeneous nucleation?	Yes. It is a containerless technique. We have used the purest material. The sample undercooled more than 300 K! At some time nucleation is inevitable, and it has to be homogeneous then. This is also indicated by the asymmetric shape of the nucleation

	statistics curve, with a steep side to higher undercoolings, showing that the undercooling temperature achieved by us is most probably the maximum undercooling achievable (in the extreme limit almost all solidification events would happen within a very small temperature interval around the ideal undercooling value of homogeneous nucleation). The reason why nucleation at the surface is preferred also for homogeneous nucleation can be explained as follows. The surface is just colder than the bulk of the droplet due to thermal losses from heat radiation. If this is not sufficient one could furthermore argue that the surface is anisotropic, curved, a defect state, necessarily with higher energy than the bulk.
Consequently, I am not convinced by the revised manuscript and the response letter. The paper gives me the feeling that the authors stretched too much from the results told by their experiments.	Yes, we took the liberty to make far-reaching conclusions of our experimental results, because viewing them together in this way gives a consistent picture of a process no one on earth is able to observe directly, not even with in situ methods.
The key conclusion cannot stand without the TEM image verifying the existence of the quasicrystalline cores.	We delivered a state-of-the-art atomic resolution HAADF-STEM which proves our twin boundary and by extrapolation of the twin boundary to a center the (ideal) structure of the tenfold twins. We do not see that there exists a better and more direct proof of our claims than that.
Thus, I cannot support the paper for publication in the current form.	We persist in our claims, which we have tried to explain in the comments above to be well funded based on facts.

REVIEWERS' COMMENTS:

Reviewer #1 (Remarks to the Author):

With the latest version, I think this comes down to an editorial decision. The authors do acknowledge that some of their conclusions are speculative, which is fine as long as the speculation is clearly differentiated from observations. Nucleation and growth in highly undercooled liquids has been and remains a highly controversial and complicated topic. Phase selection and growth are dependent on a number of complex factors, including minor impurities in addition to the degree of undercooling. While I still disagree with some of the more speculative conclusions, most of their arguments are sound. I'm not concerned about the lack of a clear QC seed as reviewer #3. This direct form of evidence would truly be difficult to find.

I would argue though that glass formability is a bit of a 'red herring' here. Many intermetallics will form glasses if cooled fast enough. The question should be are QC seeds more energetically favored in this composition range in the undercooled liquid? Or more specifically, in the undercooled range examined here. That is a question though for another study.

Reviewer #3 (Remarks to the Author):

I appreciate the efforts made by the authors in revising the manuscript. Frankly I am not fully convinced by the authors. However, considering the paper has been back and forth for several runs, I agree to proceed to publish the paper in Nature Communications. I do encourage the authors to choose opting in to the publication of the peer review files, which will help the community to get better understanding of this work.

Point-by-point response to the remarks of referees #1 and #3

Remarks of Referee #1	Comments to Referee #1
With the latest version, I think this comes down to an editorial decision. The authors do acknowledge that some of their conclusions are speculative, which is fine as long as the speculation is clearly differentiated from observations. Nucleation and growth in highly under cooled liquids has been and remains a highly controversial and complicated topic. Phase selection and growth are dependent on a number of complex factors, including minor impurities in addition to the degree of undercooling. While I still disagree with some of the more speculative conclusions, most of their arguments are sound. I'm not concerned about the lack of a clear QC seed as reviewer #3. This direct form of evidence would truly be difficult to find. I would argue though that glass formability is a bit of a 'red herring' here. Many intermetallics will form glasses if cooled fast enough. The question should be are QC seeds more energetically favored in this composition range in the undercooled liquid? Or more specifically, in the under cooled range examined here. That is a question though for another study.	We agree with the referee that we pushed the interpretation of our experimental results in a way to its limit, but we think that these more speculative conclusions are a corollary to our ideal atomistic model. Almost certainly the real structure will be different, due to the same kind of real structure defects, that are commonly found in 'ideal crystals'. However, these aberrations between the ideal model and the real crystal do not impede the usefulness of ideal models. We welcome any attempt to use our model as a starting point for a more refined treatment of the problems of nucleation and microstructure formation in the solidification of these alloys. While the question, if quasicrystalline seeds are more energetically favored, cannot, by now, be answered for NiZr, a similar question was investigated, and answered in the affirmative, by one of our colleagues for well known quasicrystalline systems some years ago (Holland-Moritz et al. Phys. Rev. Lett. 71 (1993) 1196–1199). In this work he concluded 'The results indicate that the activation energy for the formation of the nuclei of quasicrystalline phases is lower than that of crystalline phases'.
Remarks of Referee #3	Comments to Referee #3
I appreciate the efforts made by the authors in revising the manuscript. Frankly I am not fully convinced by the authors. However, considering the paper has been back and forth for several runs, I agree to proceed to publish the paper in Nature Communications. I do encourage the authors to choose opting in to the publication of the peer review files, which will help the community to get better understanding of this work.	We thank the referee for the important questions he raised during the course of the review process and agree with him that the publication of the peer review files will contribute a good deal to the understanding of the problems involved in the study of nucleation and solidification phenomena in alloy systems.